# Efficient Lung Ultrasound Classification

**DOI:** 10.3390/bioengineering10050555

**Published:** 2023-05-05

**Authors:** Antonio Bruno, Giacomo Ignesti, Ovidio Salvetti, Davide Moroni, Massimo Martinelli

**Affiliations:** Institute of Information Science and Technologies, National Research Council, 56124 Pisa, Italy; antonio.bruno@isti.cnr.it (A.B.); giacomo.ignesti@isti.cnr.it (G.I.); ovidio.salvetti@isti.cnr.it (O.S.); davide.moroni@isti.cnr.it (D.M.)

**Keywords:** convolutional neural networks, EfficientNet, lung ultrasound, SARS-CoV-2, COVID-19, pneumonia, ensemble, computer vision, supervised learning, deep learning

## Abstract

A machine learning method for classifying lung ultrasound is proposed here to provide a point of care tool for supporting a safe, fast, and accurate diagnosis that can also be useful during a pandemic such as SARS-CoV-2. Given the advantages (e.g., safety, speed, portability, cost-effectiveness) provided by the ultrasound technology over other examinations (e.g., X-ray, computer tomography, magnetic resonance imaging), our method was validated on the largest public lung ultrasound dataset. Focusing on both accuracy and efficiency, our solution is based on an efficient adaptive ensembling of two EfficientNet-b0 models reaching 100% of accuracy, which, to our knowledge, outperforms the previous state-of-the-art models by at least 5%. The complexity is restrained by adopting specific design choices: ensembling with an adaptive combination layer, ensembling performed on the deep features, and minimal ensemble using two weak models only. In this way, the number of parameters has the same order of magnitude of a single EfficientNet-b0 and the computational cost (FLOPs) is reduced at least by 20%, doubled by parallelization. Moreover, a visual analysis of the saliency maps on sample images of all the classes of the dataset reveals where an inaccurate weak model focuses its attention versus an accurate one.

## 1. Introduction

Artificial intelligence (AI), and specifically computer vision (CV), are having remarkable developments in recent years, allowing software programs to obtain meaningful information from digital images. Medicine is an area in which the experimentation and use of this technology is experiencing a strong growth [1,2,3]. Moreover, considering that only in 2020, in the United States of America alone, a production of 600 million medical images was reported [4] and that this number seems to grow steadily, it is increasingly necessary to process these data using robust and trustworthy algorithms, developed in strong collaboration at different levels among medical staff, engineers, and physics. The SARS-CoV-2 pandemic has made a quick and safe as well as economic response even more necessary: the use of point-of-care ultrasound (POCUS) to detect SARS-CoV-2 (viral) pneumonia and the bacterial one is one of the most peculiar emerging case studies which involves the use of sonography examinations in loco instead of at a dedicated facility [5,6]. In general, the preferred methods to assess pulmonary diseases are both X-ray and computed tomography (CT) due to their high image quality and diagnostic power. Nonetheless, ultrasound appears to detect signs of lung diseases as well as, or even better than, CT [7,8,9,10,11]. In some cases, there is a directed map between evidence found by ultrasound and CT [12], too. The needs requested by the SARS-CoV-2 pandemic moved the attention from a precision diagnostic approach to one that aims to maximize a trade-off between accuracy, time, and safety, which are crucial aspects in “a real emergency situation”. Physicians and researchers around the world adapted their points of view towards a common direction where the employment of lung POCUS seemed an optimal solution for both quarantined and hospitalized subjects [8,13,14]. Obviously, CT and magnetic resonance imaging (MRI) are far more precise and reliable examinations, but both have downsides that cannot be ignored. Specifically for CT, a notable downside is the associated ionizing radiation. The equivalent dose for a chest CT is around 7 mSV [15], which is roughly three times the medium annual background radiation exposure.

A clear limitation for MRI is the fact that subjects with different kinds of implants may not undergo the examination.

Moreover, both examinations are more time-consuming, much more expensive, and affected by subject cooperation. In the case of POCUS, the trade-off between pros and cons is hugely in favor of the former: it is portable, safe, cheap, and repeatable in many medical activities. As a downside, the generated images are noisy and are deeply affected by operator experience, therefore its interpretation can be less obvious than CT or MRI. Artificial intelligence can be used to narrow this gap: before 2019, deep learning (DL) methods were effectively used on ultrasound datasets [16,17], but not specifically on lung disease detection. Following the SARS-CoV-2 pandemic, an increase in articles about this topic can be noted. The underlying reasons are evident: prompt response, economic needs, scientific purposes, and an amount of data not available in the past.

### 1.1. Related Works

Two very recent and exhaustive literature reviews [18,19] highlight how much SARS-CoV-2 has sped up research on the use of lung ultrasound (LUS) and machine learning (ML). From these studies reporting several lines of research carried out, it emerged that ML and DL algorithms are mainly used for classification and segmentation tasks.

This trend can even be spotted in the first two recognized works on the application of DL in the analysis of LUS. The works of [20,21] were mainly based on a position paper that proposes LUS as a standardized approach to assessing the condition of COVID-19 patients. The work in [9] defines a standard procedure to acquire the images and proposes a score to assert the severity of the pathology. The suggested score is based on different ultrasound signs that are used as targets by different classification and segmentation algorithms. In this context, [21] built a closed-access database named Italian COVID-19 Lung Ultrasound DataBase (ICLUS-DB), while [20] mainly used open-access images to build the POCUS dataset, which at present is the most used in literature for classification. Classification studies focus primarily on the use of AI to discriminate between chest ultrasound images of healthy and pneumonia-afflicted subjects [22]. The vast majority of these studies investigate the possibility to distinguish between SARS-CoV-2 and bacterial pneumonia [23].

Among the best-performing ones, [11] employs a transfer learning approach by pretraining a network using a more consistent lung CT/MRI dataset.

The ICLUS-DB database [21] is used to investigate both classification and segmentation approaches.

Segmentation approaches studies focus more on the detection of biomarkers [21] and signs that are important in POCUS diagnosis (e.g., A-lines, B-lines) [24]. Most recent and interesting works provide solutions to discriminate LUS among healthy, pneumonia, and SARS-CoV-2 conditions with good results: some preimplemented models, available in most AI frameworks, have been applied on the dataset [25], such as Inception, VGG, and ResNet, while some of them apply modifications to those models to better fit the problem [1,26]. Novel ad hoc architectures have been proposed; the most famous and often used as a baseline is POCOVID-Net [27] and its improved version with attention mechanism [28]; a transformer from scratch is proposed in [29], and a network based on capsule (COVID-CAPS) is used [30]. Other works with lightweight convolutional neural networks (CNNs) defined from scratch are MINICOVID-NET [31,32] and lightweight transformer [33].

Since the classification of medical images entails decisions and actions involving human beings, it is crucial that the system is secure and understandable, that it is used as a second reading, and that the final decision must be made by a clinician. To this aim, an explanation of the results obtained by an automatic system must also be provided, and this is what they attemptedin [1,29,31,32,33], using activation saliency map tools to highlight the image region of interest (ROI) that the models focus on that affect the classification results. Indeed, the use of POCUS as a diagnostic tool is a debated topic [9,34,35].

### 1.2. Work Contribution

In this paper, we aim to introduce new methods and models for the analysis of LUS images in order to distinguish among SARS-CoV-2, pneumonia, and healthy conditions. The proposed new model, using EfficientNet-b0 [36] as a core, is based on a recent strategy for ensembling at the deep features level [37,38]. It reaches the state-of-the-art (SOTA) accuracy of 100% on a well-known public reference dataset. The proposed model is computationally efficient with a relatively low number of parameters and floating point operations (FLOPs), making it in principle applicable for real-time operation in the point-of-care scenario. As an additional contribution, the explainability of the ensemble model is investigated with a preliminary analysis of activation maps. This research contributes to the understanding of models and provides visual explanations of the obtained results.

The paper is organized as follows: in Section 2, the used dataset and the network architecture proposed are introduced, current results are reported in Section 3 and discussed in Section 4, while Section 5 ends the paper with conclusions and ideas for future work.

## 2. Materials and Methods

### 2.1. Dataset

The dataset used for this task is, to our knowledge, the largest publicly available LUS dataset [39], comprising a total of 261 ultrasound videos and images from 216 different patients among 41 different sources. The data were collected, cleaned, and reviewed by medical experts. In particular, for the sake of this work and comparability with the SOTA, we used the frame-based version when every single frame of each video is classified. In Table 1, data distribution is described, and in Figure 1, some examples for each class are shown. More details about the whole dataset (e.g., patient distribution, acquisition technique, sources) are described both in [1] and in the GitHub repository of the project.

### 2.2. Validation Pipeline

In this section, the validation pipeline used is motivated, discussed, and detailed. As already mentioned, the proposed method is based on a recent ensembling strategy that was presented in [37,38], where its advantages are demonstrated on a wide group of benchmark datasets for image classification. While we refer the reader to those papers for a complete treatment, we briefly review the main motivations and concepts. Ensembling is a well-known technique that combines several models, called *weak* learners, in order to produce a model with better performance than any of the weak learners alone [40]. Usually, the combination is accomplished by aggregating the output of the weak learners, generally by voting (respective averaging) for classification (respective regression). Other aspects, such as ensemble size (i.e., the number of weak learners) and ensemble techniques (e.g., bagging, boosting, stacking), are crucial for obtaining a satisfactory result. Since it requires the training of several models, ensembles make the overall validation much more expensive, and model complexity grows at least linearly with respect to the ensemble size; moreover, ensembling is a time-consuming process, and this is the main reason preventing a more extended use in practice, especially in CV. Conversely, we revisited ensembling to exploit this tool with restrained resources (e.g., with respect to model complexity, validation time, and training time) and demonstrate its usage of a classification task of high relevance in clinical practice, providing a possible method for real-time decision support in LUS. The basic idea is to start training a set of weak models—that, in our case, are based on the EfficinetNet family, which is known to have a favorable ratio between accuracy and complexity—and then to instantiate and tune an ensemble model using a subset of the best-performing weak models, through an ad hoc designed training procedure. Namely, for each weak model, only the first layers, which correspond to feature extractor, are kept, while the final classification layers, named output module, are disregarded. A combination layer is introduced to combine all the output of the weak feature extractors. Such a combination layer is trainable through a standard procedure which is achieved by keeping the feature extractors frozen. It might be argued that the proposed ensembling strategy performs a combination at the deep feature level, which seems to be a rather innovative approach, to the best of our knowledge. In addition, the training procedure of the ensembling does not produce a relevant computational burden since most of the layers of the ensemble are kept frozen. Finally, with the overall architecture being embarrassingly parallel, moving from weak learners to ensembling does not involve a significant change in computational time. Note that, in our experiments, end-to-end training using transfer learning [41] from ImageNet pretrained models [42] is performed in order to assess and select the weak models, and then the ensemble is defined and fine-tuned. Figure 2 shows the pipeline and the architectures used in our work, and the description of details follows.

#### 2.2.1. EfficientNet

Since ultrasound has the advantage of being processed in real time, the efficiency of the processing plays a crucial role. In this work, EfficientNet-b0 [36] is used as a core model because the EfficientNet architecture family is the only one that reverted the trend to trade a small accuracy improvement with a huge complexity growth (Table 2) and, according to Table 3 and previous work [37], EfficientNet-b0 is the architecture having the best accuracy/complexity trade-off. The efficiency of this architecture is given by two main factors: (i) the inverted bottleneck MBConv (first introduced in MobileNetV2) as a core module, which expands and compresses channels, reducing the complexity of convolution; and (ii) the compound scaling by which input, width, and depth scaling are performed in conjunction, since it is observed that they are dependent.

#### 2.2.2. Hyperparameters

After a previous investigation, in order to reduce the search space, some hyperparameters have been fixed:**Input size:** Set to 512 × 512 because it is the best resolution in order to have less computational cost without losing image details.**Batch size:** Set to the maximum available using our GPU (32 GB RAM) which is 50 for end-to-end and 200 for fine-tuning.**Regularization:** To prevent overfitting, early-stopping is used, and patience is set at 10, because deep models have relatively fast convergence and they usually start overfitting early, so no more patience is needed.**Optimizer:** AdaBelief [43] with learning rate $5\cdot 10^{-4}$, betas (0.9, 0.999), eps $10^{-16}$, using weight decoupling without rectifying, to have both fast convergence and generalization.**Validation metric:** Weighted F1-score which better takes into account both errors and data imbalance.**Dataset split:** 75/10/15, respectively, for train/valid/test subsets.**Standardization:** Data are processed in order to belong to a distribution with values around the mean and unit standard deviation, improving stability and convergence of the training.**Interpolation:** Lanczos for both end-to-end and fine-tuning.

Then, the search on the following hyperparameter is performed:**Seeds:** Five seeds are used for the end-to-end training (seeds affect both subset splitting and classification layer initialization) and five seeds for ensemble of the fine-tuning (affecting the combination layer initialization only).

#### 2.2.3. Efficient Adaptive Ensemble

Ensembling is the way of combining two or more models (i.e., weak models) in order to have a new combined model better than the weak ones [44]. In this work, ensembling is performed in this way:Select the best two end-to-end trained models (i.e., weak models);Remove the classification layers from the weak models;Freeze the parameters of the weak models;Initialize a fresh combination linear layer;Train the ensemble model (i.e., fine-tune the combination layer) by usual gradient descent.

This kind of ensemble is (i) efficient because only the combination layer is fine-tuned and there are just two weak models that can be executed in parallel since their processing is independent, and (ii) adaptive since the ensemble and, especially, its combination layer are trained according to the data.

## 3. Results

For the sake of robustness and comparability with the SOTA, experiments were conducted using stratified 5-fold cross-validation, keeping the frames of a single video belonging to one fold only and having the number of videos per class similar in all folds. In this way, every fold is treated as an independent task, as described in Section 2.2.

The results on the single folds reported in Table 4, Table 5, Table 6, Table 7 and Table 8 show that the ensemble obtains 100% accuracy at every run in all folds, so now we focus on end-to-end weak model results. Fold 1 (Table 4) seems to be the hardest, having no runs with 100% at every subset, and Fold 2 (Table 5) is the easiest, with 100% accuracy at every run while the remaining folds are in the “average” (see Table 9 for the mean accuracies on every subset for each fold).

Last, but not least, according to Table 10, the proposed method improves the SOTA, reducing the number of parameters and FLOPS.

## 4. Discussion

Dataset issues were extensively discussed in its official presentation paper [1]. The method presented here for classifying COVID-19, pneumonia, and healthy LUS outperforms the SOTA: its importance is in terms of the number of parameters and on complexity which are both lower than previous methods, pointing the way to an efficient and fast classification system that can be embedded in real-time scenarios. On one side, the dataset used is undoubtedly the biggest one publicly available; on the other side, it would be better to test the proposed method on even larger and more heterogeneous datasets to prove its validity. However, since such datasets are not currently available, we performed two more actions to further investigate our model: (a) progressively reducing the size of the training set without modifying the validation and test sets, and (b) progressively reducing the train set and moving the removed data to the other two sets. In both cases, deterioration is observed when there is a strong reduction (in case (a) when the train set is reduced to 40% of the original size; in case (b) when the train dataset is less than one-third of the entire dataset). Clearly, this behavior might be symptomatic of a certain overfitting level which, however, cannot be overcome, given the limits of the POCOVID dataset itself. To analyze more in-depth the model and to better understand its behavior, we applied the gradient-weighted class activation mapping (GradCAM) [45] algorithm, producing visual explanations: as the saliency maps show, our model seems to produce reasonable explanations since in all the cases it is focusingon meaningful areas. On the other side, when we apply the same method to weak models which did not obtain 100% of accuracy, it is clear that the model concentrates on areas of the LUS that are less, or not, important. Indeed, we generally noticed that an accurate classification focused on (see a representative example in Figure 3):“Evidence” usually at the upper side of the image and concentrated activations in the case of COVID-19;“Evidence” everywhere (mainly lower part) with relaxed activation in the case of pneumonia;Mainly the healthy part of the lung (black) with very expanded activation in the case of healthy.

**Figure 3 bioengineering-10-00555-f003:**
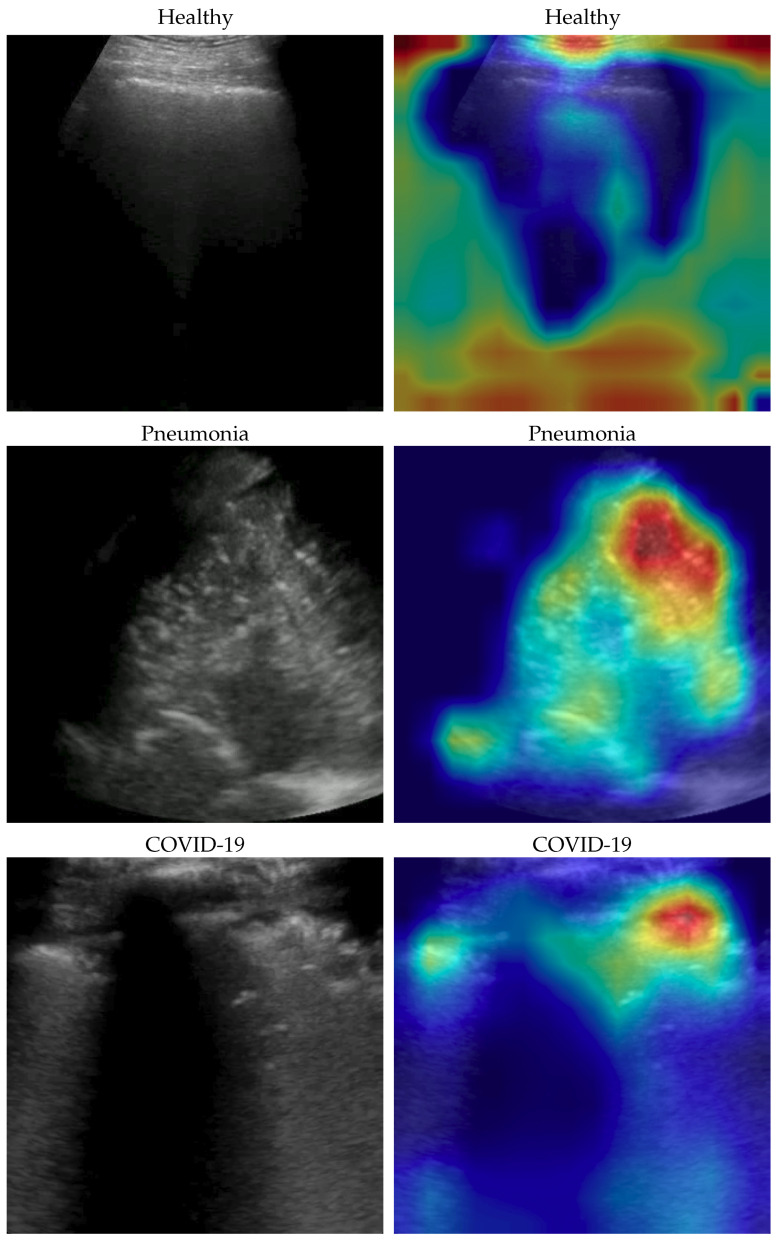
Healthy (first row), pneumonia (second row), and COVID-19 (third row) samples from the dataset and their saliency map. Red (higher) to blue (lower) scale.

Samples of saliency maps of every possible combination of wrong versus correct predictions are shown in Figure 4, Figure 5, Figure 6, Figure 7, Figure 8 and Figure 9.

From the point of view of the classification, no particular limitations can be found at the current stage; indeed, our work hugely improved (+5.3% up to +6.6%) the accuracy of the SOTA with lower complexity (in the worst case, −22% up to −61% FLOPs). As reported in Table 10, the FLOPS of the old SOTA model were not provided by the authors, but we just estimated it as lower bound; therefore, this difference may be even higher. The explainability is still an challenging open problem in DL [46,47,48]; nevertheless, our model in combination with GradCAM provides a significant support to understand the relevant information exploited to produce its output.

## 5. Conclusions

An artificial intelligence method was presented to automatically classify LUS videos into healthy, COVID-19, or pneumonia: the proposed method matches both advantages (e.g., portability, safety) and disadvantages (e.g., challenging interpretation) of LUS. An efficient adaptive ensembling model based on two EfficientNet-b0 weak models achieved an accuracy of 100% on the largest publicly available LUS dataset, improving the performance with respect to the previous SOTA, maintaining the same order of magnitude of an EfficientNet-b0 model. The extremely low computational cost makes the proposed method suitable for embedding in real-time systems; in fact, the FLOPs are comparable to values considered acceptable more than ten years ago (i.e., AlexNet, Table 2). Moreover, using saliency maps, the proposed method provides a supporting visual explanation highlighting areas of the LUS images to discern between the analyzed class.

Future investigations will focus on further improving our adaptive efficient ensembling model and applying it to classify other important signs in LUS that we are acquiring in an ongoing telemedicine project [49]. Finally, we would also like to test our work on further point-of-care analysis (e.g., A-lines, B-lines, thickness) involving object detection and segmentation tasks.

## Figures and Tables

**Figure 1 bioengineering-10-00555-f001:**
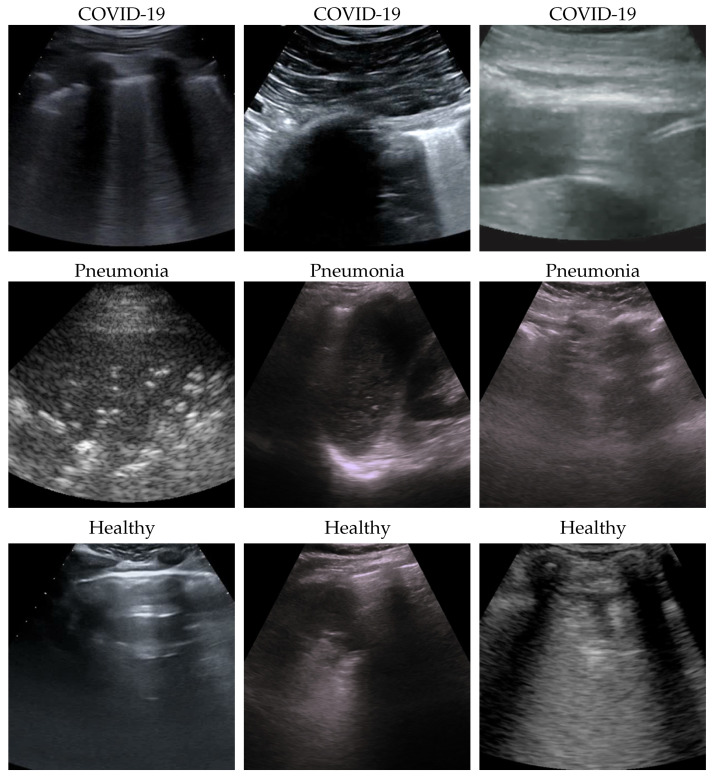
Some COVID-19 (first row), pneumonia (second row), and healthy (third row) samples from the dataset. As can be appreciated from the images, even among the same class, the sample appears to be very heterogeneous, and there is no kind of bias (e.g., pattern, color).

**Figure 2 bioengineering-10-00555-f002:**
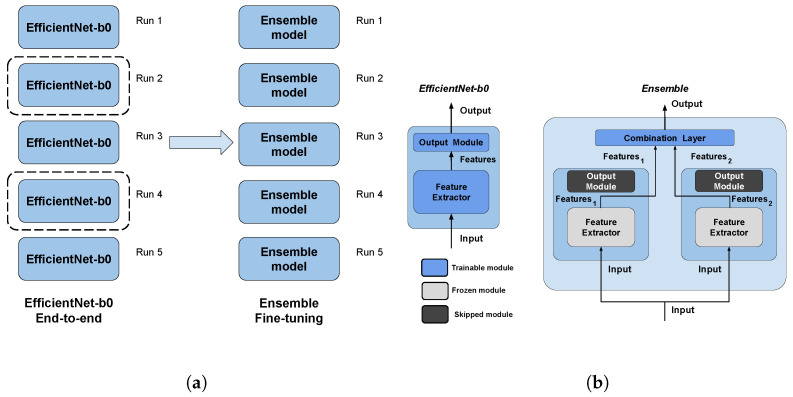
The pipeline and the architectures used in this work. The pipeline (**a**) is made by two main steps: first, end-to-end EfficienNet-b0 training, then ensemble fine-tuning using the best two models (surrounded by a dashed line) of the previous step as weak models. The architectures (**b**) used are EfficientNet-b0 for end-to-end training, and the ensemble is performed by using a trainable combination layer on the features of the weak models (dark-filled output modules are skipped); moreover, training computational complexity is reduced by freezing the parameters of weak models (light gray filled modules). Both validation steps perform runs with five different seeds (i.e., modules initialization). (**a**) Validation pipeline. (**b**) Architectures.

**Figure 4 bioengineering-10-00555-f004:**
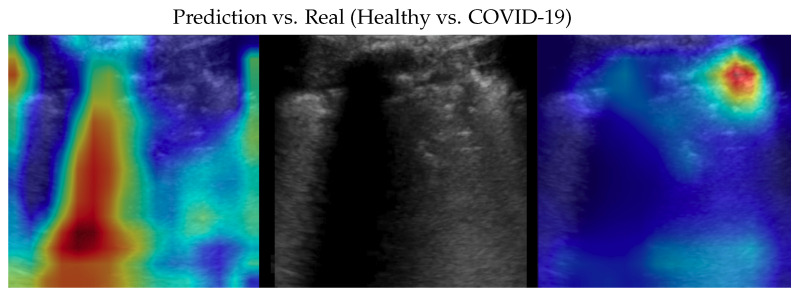
Classifications of a COVID-19 image: in the middle, the input image; on the left, the focus of a wrong classification as healthy; on the right, the focus of the correct classification obtained by the classifier achieving 100% accuracy. Red (higher) to blue (lower) scale.

**Figure 5 bioengineering-10-00555-f005:**
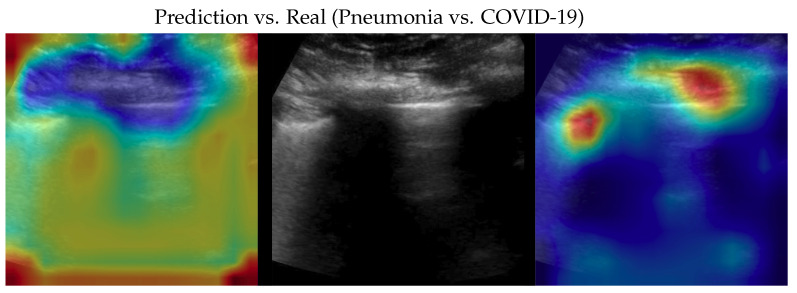
Classification of a COVID-19 image: in the middle, the input image; on the left, the focus of a wrong classification as pneumonia; on the right, the focus of the correct classification obtained by the classifier achieving 100% accuracy. Red (higher) to blue (lower) scale.

**Figure 6 bioengineering-10-00555-f006:**
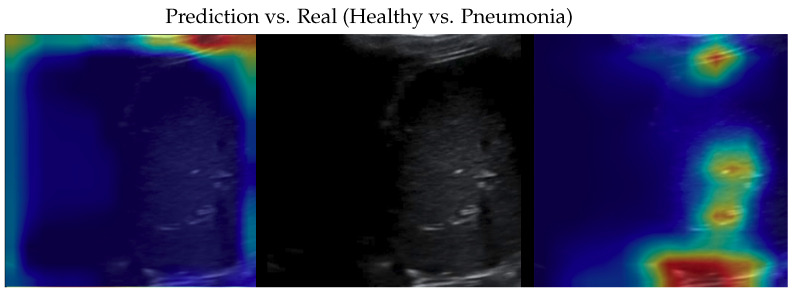
Classification of a pneumonia image: in the middle, the input image; on the left, the focus of a wrong classification as healthy; on the right, the focus of the correct classification obtained by the classifier achieving 100% accuracy. Red (higher) to blue (lower) scale.

**Figure 7 bioengineering-10-00555-f007:**
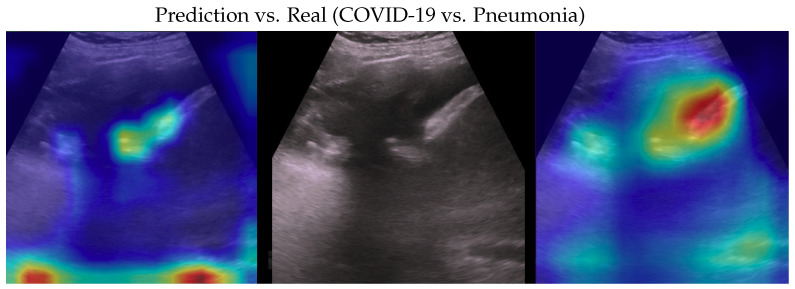
Classification of a pneumonia image: in the middle, the input image; on the left, the focus of a wrong classification as COVID-19; on the right, the focus of the correct classification obtained by the classifier achieving 100% accuracy. Red (higher) to blue (lower) scale.

**Figure 8 bioengineering-10-00555-f008:**
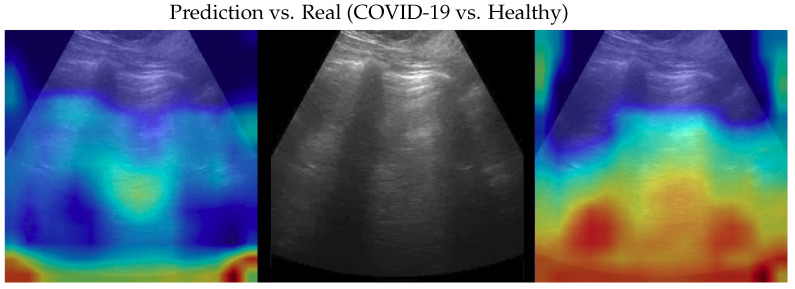
Classification of a healthy image: in the middle, the input image; on the left, the focus of a wrong classification as COVID-19; on the right, the focus of the correct classification obtained by the classifier achieving 100% accuracy. Red (higher) to blue (lower) scale.

**Figure 9 bioengineering-10-00555-f009:**
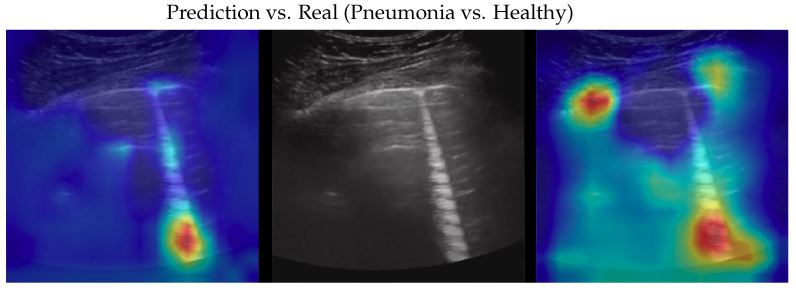
Classification of a healthy image: in the middle, the input image; on the left, the focus of a wrong classification as pneumonia; on the right, the focus of the correct classification obtained by the classifier achieving 100% accuracy. Red (higher) to blue (lower) scale.

**Table 1 bioengineering-10-00555-t001:** Dataset description with the number of videos and the corresponding number of frames for each class.

Class	Videos	Frames	Total Frames
COVID-19	70	22	1024
Pneumonia	51	22	704
Healthy	75	15	1326

**Table 2 bioengineering-10-00555-t002:** Evolution of the state-of-the-art on the ImageNet classification task. As can be seen, after EfficientNet, the complexity grows exponentially; with respect to the accuracy improvement, the same trend can be noticed on other state-of-the-art datasets (e.g., CIFAR, MNIST). N.B. Only some architectures providing relevant improvements are shown in this table.

Model	Year	Accuracy	#Parameters	#FLOPs
AlexNet	2012	63.3%	60 M	0.7 G
InceptionV3	2015	78.8%	24 M	6 G
ResNeXt-101 64 × 4	2016	80.9%	84 M	31 G
EfficientNet-b7	2019	84.3%	67 M	37 G
Swin-L	2021	87.3%	197 M	104 G
NFNet-F4+	2021	89.2%	527 M	367 G
ViT-G/14	2021	90.4%	1843 M	2859 G
ViT-e	2022	90.9%	3900 M	1980 G
CoAtNet-7 (BASIC-L)	2023	91.1%	2440 M	2586 G

**Table 3 bioengineering-10-00555-t003:** Performances of EfficientNet family on ImageNet classification task.

Model	Accuracy	#Parameters	#FLOPs
EfficientNet-b0	77.1%	5.3 M	0.39 G
EfficientNet-b1	79.1%	7.8 M	0.70 G
EfficientNet-b2	80.1%	9.2 M	1.00 G
EfficientNet-b3	81.6%	12 M	1.80 G
EfficientNet-b4	82.9%	19 M	4.20 G
EfficientNet-b5	83.6%	30 M	9.90 G
EfficientNet-b6	84.0%	43 M	19.0 G
EfficientNet-b7	84.3%	67 M	37.0 G

**Table 4 bioengineering-10-00555-t004:** Fold 1 results—this fold seems to be the hardest one, having end-to-end run with the lowest mean accuracies. With the ensemble fine-tuning, similar to the rest of the folds, 100% accuracy is reached.

Weak			Ensemble		
Test	Valid	Train	Test	Valid	Train
0.982456	1.000000	0.995536	1.000000	1.000000	1.000000
0.964912	0.927273	0.959821	1.000000	1.000000	1.000000
0.964912	0.927273	0.941964	1.000000	1.000000	1.000000
0.964912	0.927273	0.939732	1.000000	1.000000	1.000000
0.964912	0.909091	0.939732	1.000000	1.000000	1.000000

**Table 5 bioengineering-10-00555-t005:** Fold 2 results—this fold seems to be the easiest one, having 100% on all end-to-end runs; results are also confirmed by the ensemble (runs on ensemble were performed just for completeness).

Weak			Ensemble		
Test	Valid	Train	Test	Valid	Train
1.000000	1.000000	1.000000	1.000000	1.000000	1.000000
1.000000	1.000000	1.000000	1.000000	1.000000	1.000000
1.000000	1.000000	1.000000	1.000000	1.000000	1.000000
1.000000	1.000000	1.000000	1.000000	1.000000	1.000000
1.000000	1.000000	1.000000	1.000000	1.000000	1.000000

**Table 6 bioengineering-10-00555-t006:** Fold 3 results—results on this fold are unusual since the best end-to-end run obtains 100% on test and valid, but not on train, subsets. Ensemble fine-tuning confirmed reaching 100%.

Weak			Ensemble		
Test	Valid	Train	Test	Valid	Train
1.000000	1.000000	0.995402	1.000000	1.000000	1.000000
1.000000	0.981481	0.997701	1.000000	1.000000	1.000000
1.000000	0.981481	0.995402	1.000000	1.000000	1.000000
1.000000	0.981481	0.986207	1.000000	1.000000	1.000000
1.000000	0.962963	0.997701	1.000000	1.000000	1.000000

**Table 7 bioengineering-10-00555-t007:** Fold 4 results—results on this fold are “average” since some end-to-end runs obtain 100% accuracy on all subsets while the ensemble confirms 100% on all runs.

Weak			Ensemble		
Test	Valid	Train	Test	Valid	Train
1.000000	1.000000	1.000000	1.000000	1.000000	1.000000
1.000000	1.000000	1.000000	1.000000	1.000000	1.000000
1.000000	0.984127	0.996047	1.000000	1.000000	1.000000
0.984375	1.000000	0.998024	1.000000	1.000000	1.000000
0.984375	1.000000	0.996047	1.000000	1.000000	1.000000

**Table 8 bioengineering-10-00555-t008:** Fold 5 results—results on this fold are “average” since some end-to-end runs obtain 100% accuracy on all subsets while the ensemble confirms 100% on all runs.

Weak			Ensemble		
Test	Valid	Train	Test	Valid	Train
1.000000	1.000000	1.000000	1.000000	1.000000	1.000000
1.000000	1.000000	1.000000	1.000000	1.000000	1.000000
1.000000	1.000000	0.995789	1.000000	1.000000	1.000000
1.000000	1.000000	0.989474	1.000000	1.000000	1.000000
1.000000	1.000000	0.987368	1.000000	1.000000	1.000000

**Table 9 bioengineering-10-00555-t009:** End-to-end weak models training mean accuracies on every subset for each fold.

	Test	Valid	Train
Fold 1	0.968 ± 0.007	0.938 ± 0.032	0.955 ± 0.021
Fold 2	1.000 ± 0.000	1.000 ± 0.000	1.000 ± 0.000
Fold 3	1.000 ± 0.000	0.981 ± 0.012	0.994 ± 0.004
Fold 4	0.993 ± 0.008	0.996 ± 0.006	0.998 ± 0.002
Fold 5	1.000 ± 0.000	1.000 ± 0.000	0.994 ± 0.005
Average	0.992 ± 0.012	0.983 ± 0.024	**0.988 ± 0.017**

**Table 10 bioengineering-10-00555-t010:** Comparisons, with metrics for each class, of the proposed model with the SOTA. Accuracy in brackets, if any, refers to balanced accuracy. Values are reported with the same significant digits as reported in the original papers.

	Class	Recall	Precision	F1-Score
**POCOVID-Net [27]**				
Acc.: 82.1%	COVID-19	0.881 ± 0.108	0.846 ± 0.068	0.863 ± 0.083
#Param.: 14.7 M	Pneumonia	0.915 ± 0.031	0.939 ± 0.042	0.927 ± 0.028
FLOPs: 30.7 G	Healthy	0.519 ± 0.029	0.562 ± 0.082	0.540 ± 0.043
**MINICOVID-Net [31]**				
Acc.: 82.7%	COVID-19	0.918 ± 0.096	0.819 ± 0.039	0.866 ± 0.056
#Param.: 3.4 M	Pneumonia	0.903 ± 0.053	0.824 ± 0.045	0.862 ± 0.049
FLOPs: 1.15 G	Healthy	0.447 ± 0.011	0.623 ± 0.095	0.521 ± 0.010
**VGG-16 [1]**				
Acc.: 87.8% (87.1%)	COVID-19	0.88 ± 0.07	0.90 ± 0.07	0.89 ± 0.06
#Param.: 14.7 M	Pneumonia	0.90 ± 0.11	0.81 ± 0.08	0.85 ± 0.08
FLOPs: 15.3 G	Healthy	0.83 ± 0.11	0.90 ± 0.06	0.86 ± 0.08
**InceptionV3 [25]**				
Acc.: 89.1% (89.3%)	COVID-19	0.864 ± 0.036	0.901 ± 0.031	0.880 ± 0.030
#Param.: 23.9 M	Pneumonia	0.908 ± 0.025	0.842 ± 0.037	0.871 ± 0.025
FLOPs: 6 G	Healthy	0.907 ± 0.026	0.918 ± 0.021	0.911 ± 0.021
**DenseNet-201 [26]**				
Acc.: 90.4%	COVID-19	0.892	0.918	0.905
#Param.: 20 M	Pneumonia	0.903	0.610	0.728
FLOPs: 4.29 G	Healthy	0.850	0.842	0.846
**Light Transformer [33]**				
Acc.: 93.4%	COVID-19	0.958 ± 0.025	0.958 ± 0.012	0.951 ± 0.017
#Param.: 0.3 M	Pneumonia	0.948 ± 0.013	0.951 ± 0.038	0.949 ± 0.020
FLOPs: 1 G ^a^	Healthy	0.877 ± 0.034	0.912 ± 0.037	0.894 ± 0.036
**Weak model (our)**				
Acc.: 98.7% (98.3%)	COVID-19	0.984 ± 0.004	0.993 ± 0.004	0.990 ± 0.004
#Param.: 5 M	Pneumonia	0.997 ± 0.005	0.991 ± 0.006	0.991 ± 0.007
FLOPs: 0.39 G	Healthy	0.999 ± 0.003	0.993 ± 0.003	0.995 ± 0.004
**Ensemble (our)**				
Acc.: 100% (100%)	COVID-19	1.000 ± 0.000	1.000 ± 0.000	1.000 ± 0.000
#Param.: 10 M ^b^	Pneumonia	1.000 ± 0.000	1.000 ± 0.000	1.000 ± 0.000
FLOPs: 0.78 G ^c^	Healthy	1.000 ± 0.000	1.000 ± 0.000	1.000 ± 0.000

^a^ FLOPs or any computational-cost-related metrics are not provided in this work; however, even if the number of parameters is very low (due to weight sharing), the attention modules are the bottleneck of the transformer and perform a huge amount of FLOPs, and the types of attention they used usually perform at least 1GFLOPs. ^b^ The actual trainable parameters are 0.1 M (the parameters of the combination layer), and the backward pass during the training ends very early because there is no reason to propagate gradients to the input layer. ^c^ The forward pass can be parallelized, having the same execution time of a weak model.

## Data Availability

Data used in this work are publicly available at https://github.com/jannisborn/covid19_ultrasound accessed on 4 May 2023.

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
