# Peer review of "Efficient Lung Ultrasound Classification"

_bioengineering, 2023, doi:10.3390/bioengineering10050555_

Round 1
Reviewer 1 Report
This topic seems interesting and applicable. The authors could provide a well-written manuscript. However, there are some issues to be addressed at this stage based on the following comments:
1- Please polish the English of the manuscript. I found several errors.
2- Do not miss the well-known acronyms; e.g., ML.
3- You have too many keywords; besides, SARS-CoV-2; COVID-19 at the same time?
4- Extend the introduction and literature review sections by talking more about the problem significance, challenges, your contributions based on the research gaps (which requires reviewing more recent works).
5- A flowchart or schematic research framework may help the readers to learn the main ideas and novelties of your methodology.
6- More comparative and sensitivity analyses should be made to validate the superiority of your work.
7- Discuss the limitations and outlook of the study in the conclusion section.
Author Response
Response to Reviewer 1 Comments
Point 1: Please polish the English of the manuscript. I found several errors.
Response 1: Thank you for your constructive comments. We have revised the paper taking into utmost consideration all remarks raised by the reviewers as explained in detail below. A special markup has been used for all the changes made in the submitted revised version of this paper. Our manuscript was entirely revised by the authors also with the help of a proficient English speaker.
Point 2: Do not miss the well-known acronyms; e.g., ML.
Response 2: we double-checked and extended all the abbreviations.
Point 3: You have too many keywords; besides, SARS-CoV-2; COVID-19 at the same time?
Response 3: Dear Reviewer, we followed the instructions for authors at the following address:
https://www.mdpi.com/journal/bioengineering/instructions
”Keywords: Three to ten pertinent keywords need to be added after the abstract. We recommend that the keywords are specific to the article, yet reasonably common within the subject discipline.”
As requested, we added a number of keywords in the expected range.
Regarding SARS-CoV-2 and COVID-19, we used both for indexing purposes since people use both terms: Severe Acute Respiratory Syndrome Coronavirus-2 (SARS-CoV-2) is the name given to the novel coronavirus of 2019 while COVID-19 is the name given to the disease associated with the virus.
If you think we have to remove one, we would like to keep COVID-19 because it is the most commonly used.
Point 4: Extend the introduction and literature review sections by talking more about the problem significance, challenges, your contributions based on the research gaps (which requires reviewing more recent works).
Response 4: Thank you. We extended the related works section with nine more references (e.g POCOVID-Net, MINICOVID--net, Transformer-based) that cover both architectural and explainability aspects.
Point 5: A flowchart or schematic research framework may help the readers to learn the main ideas and novelties of your methodology.
Response 5: Thank you for your comment: in Materials and Methods we added Figure 2 to show the validation pipeline steps, while we extended the 2.2 section, and each novelty is discussed in the related paragraph (2.2.x).
Point 6: More comparative and sensitivity analyses should be made to validate the superiority of your work.
Response 6: Dear Reviewer, in the paper we used seven metrics: accuracy, balanced accuracy, number of parameters, FLOPS, recall, precision and F1. In all cases, our work is the one achieving the best performance. Those are the metrics available to compare with previous works which used the same metrics. In any case, if you have any suggestions for further metrics, please let us know and we will try to incorporate additional metrics.
Point 7: Discuss the limitations and outlook of the study in the conclusion section.
Response 7: Thank you for your comment. At the current stage, we see no particular limitations on the aspect of classification, indeed our work hugely improved (+5.3 up to 6.6%) the accuracy of the SOTA and reduced the complexity (-22% up to -61% FLOPs) on the POCUS dataset. However, there are limitations in the full explainability of the results. Indeed, our method, like every deep learning algorithm, cannot be fully explainable (correlation is not causation); we addressed this issue by using an explainability tool that highlights the regions in the image that the model considers relevant to perform its choice, as shown in Figs. 3-9 and discussed in Sec. 5.
In addition, notice that we tested our method on a dataset that is the most complete and a de facto benchmark, although it might be of course extendable. Finally, in order to improve the discussion, we extended the Discussion section by adding further studies we would like to carry out, that is, testing our work on detection/segmentation tasks on further point-of-care analysis or other domains.
Reviewer 2 Report
Dear Editor,
I recommend this paper with minor edits. Hence, I request the authors to update my following comments.
1. Insert the related work as new section (especially the lung cancer classification).
2. In Section 2.1 dataset, you have mentioned 261 patients but in Table 1 there is only 169 samples (64+49+66).
3. In the abstract, mentioned as "The complexity of this solution keeps the number of 8 parameters in the same order as an EfficientNet-b0 by adopting specific design choices that are 9 adaptive ensembling with a combination layer, ensembling performed on the deep features, minimal 10 ensemble only two weak models."
In the aforementioned statement ""Only two weak models" is not clear. In other words, split the sentence in half. Include or mention the two weak models as well.
4. In Table 8, the author has specified "# param as 10M2" whereas other models the number of parameters are less. Mention the main ideas and benefits of utilizing your model's 10M2 parameters.
5. In Conclusion section, you have mentioned "the proposed method matches both advantages 178 (e.g. portability, safety) and disadvantages (e.g. challenging interpretation) of LUS."
How does the portability of the suggested method compare to your 10M2 parameter?
Author Response
Response to Reviewer 2 Comments
Point 1: Insert the related work as new section (especially the lung cancer classification).
Response 1: Thank you for your constructive comments. We have revised the paper and refactored the introduction moving the related works to a specific section. Notice however that our paper focuses on the use of ultrasound imaging which is not a reference diagnostic imaging for dealing with lung cancer. We didn’t mention lung cancer in our work, we think you meant LUS classification, maybe? If not, thank you for providing us with your idea: it could be an exploratory topic to see how our method can also face this topic.
Point 2: In Section 2.1 dataset, you have mentioned 261 patients but in Table 1 there are only 169 samples (64+49+66).
Response 2: Thank you for your comment. We missed some videos in the table and we added them (we used all during the experimentation anyway). The patients are 216 (we mistyped it) while 261 is the number of videos + single images, it means that a patient may be represented, in the extreme case, by one single image only. However, as we said in the paper, the reference [1] provides more detailed information (in this particular case, see its Table 1).
Point 3: In the abstract, mentioned as "The complexity of this solution keeps the number of 8 parameters in the same order as an EfficientNet-b0 by adopting specific design choices that are 9 adaptive ensembling with a combination layer, ensembling performed on the deep features, minimal 10 ensemble only two weak models." In the aforementioned statement ""Only two weak models" is not clear. In other words, split the sentence in half. Include or mention the two weak models as well.
Response 3: Thank you. We split and reorganized the aforementioned sentence which was not clear.
Point 4: In Table 8, the author has specified "# param as 10M2" whereas other models the number of parameters are less. Mention the main ideas and benefits of utilizing your model's 10M2 parameters.
Response 4: In the previous version of the table, the other models, except one weak model with 5M (still proposed by us, have more parameters (14.7M, 20M and 23.9M) and the difference is even more when considering the FLOPs (that is the actual processing “time”). As said in the notes “2” and “3” (“b” and “c” in the revised version, respectively), the model with 10M parameters, since it is “embarrassingly parallel”, has the same computational cost as a single weak model (having half parameters and FLOPs) but +2.3% in accuracy and improvement in every metric. During the revision we added more models to compare to, finding a method with much fewer parameters but our method still remains the one with the best accuracy and least FLOPS. Notice that in “10M2” the number 2 was the index of the footnote; we agree that this might lead to possible confusion and we have used a different numbering changing 1,2,3, into a,b,c,...
Point 5: In Conclusion section, you have mentioned "the proposed method matches both advantages 178 (e.g. portability, safety) and disadvantages (e.g. challenging interpretation) of LUS."
How does the portability of the suggested method compare to your 10M2 parameter?
Response 5: Thank you for your comment. We revised the Conclusion, explaining better that our work improves the interpretation of the LUS giving excellent results exploiting the portability of the LUS since the extremely low computational cost makes the proposed method suitable for embedding in real-time systems. Indeed, it can give “real-time” output due to low computational complexity (since it is comparable with models whose complexity was considered “acceptable” in 2014-2015, e.g. with AlexNet/ResNet, Table 2). Concerning the number of parameters more details have been introduced as already detailed in Response 4 above.
Reviewer 3 Report
1. Introduction should be carefully rewritten. The motivation should be better elaborated.
2. The authors should highlight the main novelty of the proposed method.
3.The study lacks a theoretical framework which is important for the reader to grasp the crust of the research.
4. Explain why the current method was selected for the study, its importance and compare with traditional methods.
5. Improve the quality of figures and explain those properly.
6. The language usage throughout this paper need to be improved, the author should do some proofreading on it.
7. The literature review is far from enough. It is not articulated why the existing work is not enough.
8. An Abstract, there is no brief discussion of research motivation, potential issues, and how the proposed work resolve the issue.
9. The research results reported are too premature for publication. More work is needed to substantiate the conclusions in your manuscript.
10. Does this kind of study have never attempted before? Justify this statement and give an appropriate explanation to do so in this paper.
11. Author is advised to incorporate the recent research work published in this area. Compare and contrast your work with the latest algorithms and principles developed in the past three years. (at least three methods)
Author Response
Response to Reviewer 3 Comments
Point 1: Introduction should be carefully rewritten. The motivation should be better elaborated.
Response 1: Thank you for your constructive comments. We have revised the paper taking into utmost consideration all remarks raised by the reviewers as explained in detail below. A special markup has been used for all the changes made in the submitted revised version of this paper. In particular, the introduction has been reworked and extended, making more clear context and motivation, related works and paper contribution.
Point 2: The authors should highlight the main novelty of the proposed method.
Response 2: Thank you for your comment: in Materials and Methods we added Figure 2 to show the validation pipeline steps, while we extended the 2.2 section, and each novelty is discussed in the related paragraph (2.2.x). In addition, notice that some of the classification algorithms that were used to analyze the POCOVID dataset (the de facto benchmark datasets addressed in our study) are considered overcome by the machine learning and deep learning community. Indeed some attempts in the literature (to which we compare) use algorithms like AlexNet (2012) and VGG (2014). Such algorithms are key literature and were designed specifically for the classification task of ImageNet, but in the latter years, their structure was regarded as suboptimal. The fixed dimension of the convolutional filter (3x3 for VGG) creates an oversaturation of parameters that feed in the fully-connected layer without fully representing all the image details. Successive approaches like Inception and ResNet propose a solution to this problem by introducing concepts such as different kernel size operations, batch normalization and residual connection that fairly upgrade the network. After this other important work are MobileNet and DenseNet, both published in 2017; these two algorithms respectively propose an optimization for reducing the number of training parameters without losing accuracy while DenseNet explores the importance of the connection between layers of the network to augment robustness. EfficientNet (2019) uses all these concepts to build a network with a minimum set of parameters and decent accuracy. In our study we used EfficinetNet has a basic architecture and then obtained a significant improvement thanks to a quite recent strategy for ensembling as described in Response 3.
Point 3: The study lacks a theoretical framework which is important for the reader to grasp the crust of the research.
Response 3: Thank you for your comment. Indeed, this work is based on a general theoretical framework centred on a new strategy for ensembling that has been discussed by us in a preliminary workshop paper and in a full paper. Such papers are cited in the main text as references:
- Bruno, A.; Moroni, D.; Martinelli, M. Efficient Adaptive Ensembling for Image Classification. arXiv preprint arXiv:2206.07394, 2022
- Bruno, A.; Martinelli, M.; Moroni, D. Exploring Ensembling in Deep Learning. Pattern Recognition and Image Analysis 2022, 32, 519–521.
However, we agree that such a point was not sufficiently stressed and, for the sake of completeness, we reworked the paper adding more details about the theoretical framework. This has been done by introducing the new subsection “work contribution”, a new Figure with the overall methodology (Figure 2) and extending the text in Section 2.2.
Point 4: Explain why the current method was selected for the study, its importance and compare with traditional methods.
Response 4: Thank you for your comment. As explained in the previous points, we have now revised the text adding more details on the general framework, on the selection of the basic “weak” learners' architectures and on the ensembling strategy. An optimal trade-off between accuracy and complexity was sought; for this reason, we have reported the performance of the members of the EfficinetNet family whose representatives are used as weak learners in our study. Further, we added a more complete comparison with more methods in the revised Table 10 in the paper. We also stressed the importance of being able to perform image classification with constrained resources since it might open new opportunities in the real-time processing of LUS in point-of-care scenarios.
Point 5: Improve the quality of figures and explain those properly.
Response 5: We reworked all the captions improving their clarity. All the images, even the original ones from the dataset, are linked to the submitted latex document, their resizing is caused by the page layout: they all are of 300 DPI, the size of the smallest ones (that is, Figure 3 to 8) is 4056 x 1309 pixels.
Point 6: The language usage throughout this paper need to be improved, the author should do some proofreading on it.
Response 6: Thank you for your comment, our manuscript was entirely revised by the authors also with the help of a proficient English speaker.
Point 7: The literature review is far from enough. It is not articulated why the existing work is not enough.
Response 7: Thank you. We have extended and articulated the literature review with nine more references (e.g POCOVID-Net, MINICOVID--net, Transformer-based) that cover both architectural and explainability aspects.
Point 8: An Abstract, there is no brief discussion of research motivation, potential issues, and how the proposed work resolve the issue.
Response 8: Thank you. We revised the Abstract trying to keep in mind the suggestion of the reviewer. Motivation has been identified in the opportunity of having an expert system for facing diagnostic problems in a point-of-care scenario where results’ accuracy and speed are the crucial factors. The state-of-the-art only provides suboptimal image classifiers (which is an issue) while the proposed research resolves this issue by proposing a new method for image classification which is at the same time more accurate and with restrained complexity, making it suitable for point-of-care applications. We hope that the new abstract, therefore, shows the research motivation, the issue to be solved and the way it is addressed by our solution.
Point 9: The research results reported are too premature for publication. More work is needed to substantiate the conclusions in your manuscript.
Response 9: Pneumonia and Covid signs deep learning classification at the same time is indeed a new topic and literature is scarce, nevertheless this work is based on previously validated research and the feedback by the medical community on the topic is positive; therefore we would like to underline that our result is meaningful and of interest for translational medicine. We also believe that our results, which are of course built onto an existing corpus of literature, are sufficiently mature. Indeed, the method exploits a recent ensembling strategy that has been already proven successful in a number of general benchmark datasets of academic relevance (see Response 3). In this paper, the capabilities of our method in dealing with a dataset of clinical relevance have been proven. On this dataset we obtained accuracy results superior to state of the art with fewer FLOPs, also proving that such accuracy is achievable with constrained resources.We added more works we compared to.
Point 10: Does this kind of study have never attempted before? Justify this statement and give an appropriate explanation to do so in this paper.
Response 10: Other works literally took pre-implemented models from the most famous deep learning frameworks and applied them to the POCOVID dataset. By converse, we adopted specific design choices to deal with potential issues in the 2.2.x sections, moreover we used an ensemble strategy. Ensembling is generally known as a powerful tool to improve results but we stress that we obtained an ensemble while restraining the complexity and by using it in an adaptive way (usually, ensemble is done by a “fixed” aggregation function). Sometimes, ensemble, especially in computer vision, is not used for the following main reasons:
- the computation time of the ensemble grows at least linearly in the number of weak models;
- the validation search space grows exponentially due to the new hyperparameters introduced by the ensemble;
- baseline computer vision models alone with high accuracy are hugely complex (>1000M parameters and >1000G FLOPs, we added a new table, now Table 1, that summarizes the trend)
By converse our strategy does not suffer from these limits as explained in Section 2.
Point 11: Author is advised to incorporate the recent research work published in this area. Compare and contrast your work with the latest algorithms and principles developed in the past three years. (at least three methods)
Response 11: Thank you for your comment. We updated our article by adding a comparison with the results of three additional algorithms, comparing in total against six algorithms (Table 10).
Reviewer 4 Report
In the manuscript, the authors proposed to use an efficient adaptive ensembling of two EfficientNet-b0 models to realize an accurate diagnosis based on lung ultrasound. Although the results shown are very strong, I feel the manuscript is somehow suspicious:
1. All five authors contributed equally to this work? And 10 key words?
2. For this dataset, there are only 3 categories of labels, yet the authors used 2 EfficientNet-b0 models and called each of them an inaccurate weak model. However, on most folds, the performance of one weak model is the same, or very close to the ensemble model.
3. On some folds, the weak model shows a validating and testing performance even better than the training performance. And the ensemble model shows 100% accuracy for training/validating/training datasets for all folds.
At this stage, I cannot recommend the manuscript for publication in the journal.
Author Response
Response to Reviewer 4 Comments
In the manuscript, the authors proposed to use an efficient adaptive ensembling of two EfficientNet-b0 models to realize an accurate diagnosis based on lung ultrasound. Although the results shown are very strong, I feel the manuscript is somehow suspicious.
Thank you for your comments. We have revised the paper taking into utmost consideration all remarks raised by the four reviewers as explained in detail below. A special markup has been used for all the changes made in the submitted revised version of this paper. We sincerely hope that the revised version might allay previous doubts about the paper.
Point 1: All five authors contributed equally to this work? And 10 key words?
Response 1: We specified more in detail in which way the authors contributed to this work in the section Author Contributions; all the authors have agreed that there is substantial parity in the contribution given. As for the keywords, MDPI bioengineering guidelines suggested we cite up to 10 keywords, and we indicated the most relevant ones to favor better indexing of the paper (which is important to foster a good impact of the paper making it findable by query in Google scholar, Scopus and other databases of relevance). Still, if you suggest a selection of the keywords we might consider changing them. Notice that we included both SARS-CoV-2 and COVID-19 since the first one is used mainly in scientific production in the clinical domain, while COVID-19 is more common in general academic production.
Point 2: For this dataset, there are only 3 categories of labels, yet the authors used 2 EfficientNet-b0 models and called each of them an inaccurate weak model. However, on most folds, the performance of one weak model is the same, or very close to the ensemble model.
Response 2: In an ensemble, the number of weak models is not related to the number of output classes, moreover, usually more weak models are used and better results are achieved (since more “knowledge” is used), at the expense of complexity that grows. We used the minimal number of weak models (i.e. 2) obtaining better results by restraining the complexity. An ensemble doesn’t overkill a task having “only” 3 classes because there is no correlation between the number of classes and the complexity of a task. Indeed in the literature, there exist binary tasks much harder than multiclass ones.
Point 3: On some folds, the weak model shows a validating and testing performance even better than the training performance. And the ensemble model shows 100% accuracy for training/validating/training datasets for all folds.
Response 3: It can happen depending on the data distribution, moreover, during training, the dropout of the EfficientNet adds a kind of “noise” to the data that may lead to a bit lower training accuracy improving validation and test.
Round 2
Reviewer 1 Report
Good work. The authors could address my comments well.
Author Response
Thank you. We made some efforts in addressing all the reviewers comments
Reviewer 3 Report
In the revised version of the manuscript, all my comments were considered. I thank the authors for that and appreciate their effort.
Author Response

(The authors gave the same response as above.)

Reviewer 4 Report
N/A
Author Response

(The authors gave the same response as above.)
